

# Development of LAMP assay for early detection of *Yersinia ruckeri* in aquaculture

Hoda Abbas, Nickala Best, Gemma Zerna and Travis Beddoe

Department of Animal, Plant and Soil Science, La Trobe University, Bundoora, Victoria, Australia

## ABSTRACT

*Yersinia ruckeri* is the causative agent of yersiniosis or enteric red mouth disease (ERM) that causes significant economic losses in the salmonid aquaculture industry. Due to an increasing number of outbreaks, lack of effective vaccines and the bacteria's ability to survive in the environment for long periods, there is a necessity for novel measures to control ERM. New techniques capable of rapidly detecting *Y. ruckeri* are critical to aid effective control programs. Molecular methods, like real-time polymerase chain reaction, can detect *Y. ruckeri*; however, that methodology is not field-deployable and cannot support local decision-making during an outbreak. We present a field-deployable molecular assay using loop mediated isothermal amplification (LAMP) and water filtering method for the detection of *Y. ruckeri* eDNA from water samples to improve current surveillance methods. The assay was optimised to amplify the glutamine synthetase gene (*glnA*) of *Y. ruckeri* in under 20 min. The assay demonstrated high specificity and sensitivity, as it did not amplify any non-target bacteria typically present in water sources. It achieved a limit of detection (LOD) of $0.5 \times 10^{-7}$ ng/μl, significantly surpassing the LOD of $0.5 \times 10^{-4}$ ng/μl obtained through conventional polymerase chain reaction (cPCR). When applied to environmental water samples spiked with transformed *Escherichia coli* containing the G-block of the *Yersinia ruckeri* (*glnA*) target gene, the *Yr*-LAMP method exhibited an analytical sensitivity of 0.08 cells/μl from the initial filtered water sample. Notably, the cumulative time for sample preparation and amplification was under 1 h. The simplicity of the developed field-deployable *Yr*-LAMP assay makes it suitable as a routine procedure to monitor fish for ERM infection. This will enable informed decision-making on mitigating pathogen prevalence in aquaculture farms.

## INTRODUCTION

Yersiniosis is a highly infectious disease causing significant economic losses in fish aquaculture worldwide, especially in salmonids (*Keeling et al., 2012*; *Oroji et al., 2021*; *Yang et al., 2024*). *Yersinia ruckeri* is the etiological agent of enteric red mouth disease (ERM) or Yersiniosis (*Keeling et al., 2012*). *Y. ruckeri* is an anaerobic, gram-negative, rod-shaped bacteria that mainly enters the host body through the pavement cells of the gill lamellae,

Corresponding author
Travis Beddoe,
t.beddoe@latrobe.edu.au

then into the intestine and bloodstream (*Keeling et al., 2012*; *Kumar et al., 2015*). *Y. ruckeri* is a facultative intracellular pathogen, beginning the infective stage in the extracellular space then inhabiting the macrophages during the intracellular phase. The pathogen can also lay dormant in some fish without causing clinical symptoms; however, the asymptomatic hosts transmit disease through defecation to more susceptible fish (*Guijarro et al., 2018*; *Kumar et al., 2015*). The common symptoms of the disease include subcutaneous haemorrhage around the mouth, throat, tongue, gills, fins and gums in addition to an enlarged spleen and haemorrhages on the liver's surface, pancreas, swim bladder or lateral muscles (*Kumar et al., 2015*). The severity of infections in fish can vary, as some fish naturally recover from infections; however mass mortality has been observed during outbreaks (*Yang et al., 2024*). While there is no precise assessment of the economic toll caused solely by *Y. ruckeri* infections, it is one of the significant pathogens in aquaculture, contributing to a combined annual loss of over USD 6 billion (*Stentiford et al., 2017*). *Y. ruckeri* is classified into two biotypes (motile and non-motile) and four major serotypes, which have different surface antigens (*Carson et al., 2008*; *Kumar et al., 2015*). Serotype O1b biotype I has predominantly been responsible for most outbreaks in the past, which led to the early development of a very effective vaccine; however, recent outbreaks in vaccinated fish caused by biotype II, has raised concerns about the ability to control *Y. ruckeri* infections (*Austin, Robertson & Austin, 2003*; *Fernández, Méndez & Guijarro, 2007*; *Guijarro et al., 2018*; *Kumar et al., 2015*; *Saleh, Soliman & El-Matbouli, 2008*). Without the availability of effective vaccines to control infection in all countries, there is an increased reliance on other control measures such as antibiotics and preventative management practices (*Önalan & Çevik, 2020*; *Valle et al., 2020*). Ideally, early pathogen detection will guide on-farm management decisions, enabling a swift response to increasing infection levels and supporting effective pathogen control.

Traditional methods for the detection of *Y. ruckeri* are bacterial cultures (*Rodgers, 1992*; *Waltman & Shotts, 1984*), biochemical tests (*Davies & Frerichs, 1989*; *De Grandis et al., 1988*) and serological tests (*Olesen, 1991*; *Stevenson & Airdrie, 1984*). Although those methods are adequate for detection, they are laborious, can require particular growth conditions, and may not be applicable to all isolates, which can be challenging to differentiate between closely related species (*Austin, 2019*; *Davies & Frerichs, 1989*; *Ibrahim et al., 1993*). Several molecular techniques, such as restriction fragmentation-length polymorphism (RFLP) (*Garcia et al., 1998*), polymerase chain reaction (PCR) (*Gibello et al., 1999*; *LeJeune & Rurangirwa, 2000*) and real-time PCR (*Birkenbach, 2022*; *Ghosh et al., 2018*; *Keeling et al., 2012*) can successfully detect low levels of the bacteria; however, they require expensive equipment, trained personnel, and are not suitable for point-of-care pathogen detection (*Knox, Zerna & Beddoe, 2023*).

Loop-mediated isothermal amplification (LAMP) is a method for amplifying DNA at a constant temperature. It uses four to six primers to target six to eight specific regions on a DNA template. The amplification starts with a strand-displacing DNA polymerase that initiates the synthesis of new DNA. Two of the primers form loop structures, which facilitate further rounds of amplification. One of the main benefits of LAMP is that it does not require a specialized thermocycler; instead, the reaction can take place at a stable
temperature using either a heat block or a water bath. This makes LAMP a fast, straightforward, and economical choice for field applications, while still achieving the high sensitivity and specificity associated with traditional molecular techniques (*Notomi et al., 2000*). There has been LAMP assays developed for use in the aquacultural industry, for detection of viral (*Caipang et al., 2004*; *Gunimaladevi et al., 2005*), bacterial (*Savan et al., 2004*; *Tsai et al., 2013*) and parasitic pathogens (*El-Matbouli & Soliman, 2005*; *Picón-Camacho et al., 2013*). Furthermore, there has been a LAMP assay developed for the detection of *Y. ruckeri*, termed enteric red mouth LAMP (ERM-LAMP), however this test requires extensive sample preparation and DNA extraction within a specialised laboratory (*Saleh, Soliman & El-Matbouli, 2008*). The existing ERM-LAMP assay utilizes five primers targeting the *yruI/yruR* gene of *Y. ruckeri* and can be detected after one hour of incubation at 63 °C using visual inspection, agarose gel electrophoresis or by real-time monitoring of turbidity. Its high sensitivity allows detection of as low as 10 pg of *Y. ruckeri* genomic DNA. However, the ERM-LAMP method relies on the collection of fish to carry out the initial DNA extraction using tissue samples (*Saleh, Soliman & El-Matbouli, 2008*). While this is the standard testing method, it is not ideal to sacrifice several expensive, large fish for routine surveillance, nor to require specialists for septic dissection. Therefore, using environmental water for sampling offers a safer and simpler routine point-of-care surveillance method (*Khodaparast et al., 2022*).

Here, we report the development of a *Y. ruckeri*-specific LAMP (*Yr*-LAMP) assay and subsequent DNA extraction method suitable for field use from water, to allow rapid, reliable, and robust detection of *Y. ruckeri* within 1 h.

# MATERIALS AND METHODS

## *Yersinia ruckeri* LAMP primer design

Alignment of the glutamine synthetase (*glnA*) gene sequences was performed using Clustal Omega software to find regions of similarity in all *Y. ruckeri* strains and variations if compared with other non-target species. Four primer sets, designated as Yr#1, Yr#2, Yr#3, and Yr#4, were designed to target the *glnA* gene. Yr#1 and Yr#2 LAMP primer sets were generated using Primer Explorer V5 according to the DNA sequence of the *glnA* gene of *Y. ruckeri*. Conversely, Yr#3 and Yr#4 were designed manually following the LAMP protocol developed by *Notomi et al. (2000)* (Table 1). The four primer sets included inner, FIP & BIP, outer F3 & B3, and loop primer LF, except for set Yr#2 which has two loop primers, LF & LB. The suggested primers were synthesized by Integrated DNA Technologies (IDT).

## *Yersinia* spp. synthetic DNA preparation for assay optimization

A synthetic positive and negative controls were designed for assay optimisation and validation, to cover 586, 599 and 593 base pairs (bp) of the glutamine synthetase (*glnA*) from *Y. ruckeri* (AY333067.1), *Y. rohdei* (AY333059.1) and *Y. frederiksenii* (AY333030.1) respectively were synthesized by IDT. The synthetic DNA were ligated into *pCRBlunt II-TOPO* vector using the Zero Blunt™ TOPO™ PCR plasmid kit (Invitrogen, Waltham, MA, USA) according to manufacturer's instructions and transformed in *E. coli* DH5α cells

**Table 1 LAMP primer sets designed to target the glutamine synthetase (*glnA*) gene of *Yersinia ruckeri*.**

| Primer set | Primer | Sequence (5′–3′) | Length (bp) |
|---|---|---|---|
| Yr#1 | F3 | TGTTCGGACCAGAACCTGAA | 20 |
| | B3 | GGCAGAACGCAGATCTTGC | 19 |
| | FIP | GGAGTTCCATGCGCCTTCGATAACGATATTCGCTTTGGCAGC | 42 |
| | BIP | AGGTGGTAACAAAGGCCATCGTATCAACTGGGGGAACCGG | 40 |
| | LF | AACGTGGGAGCCACGGA | 17 |
| Yr#2 | F3 | GGTGGTAACAAAGGCCATCG | 20 |
| | B3 | GCTACGTTGTGCACGACAT | 19 |
| | FIP | CATGGCAGAACGCAGATCTTGCTAAAAGGCGGTTACTTCCCG | 42 |
| | BIP | AAGCACATCACCACGAAGTCGCTCCGCTTTCTTGGTCATGG | 41 |
| | LF | GCGGAATCAACTGGGGGAA | 19 |
| | LB | CTGCTGGTCAGAACGAAGTG | 20 |
| Yr#3* | F3 | CGCAGTAAAAGGCGGTTACTT | 21 |
| | B3 | GCTACGTTGTGCACGACA | 18 |
| | FIP | TGCTTCAACAACCAGACCCATATCTCCCAGTTGATTCCGCGC | 42 |
| | BIP | TCGCCACTGCTGGTCAGAATTGGATTTCATCCGCTTTCTTGGT | 43 |
| | LF | GGTTAAACACATGGCAGAACGC | 22 |
| Yr#4* | F3 | ACCAGAACCTGAATTCTTCT | 20 |
| | B3 | GGCAGAACGCAGATCTTGC | 19 |
| | FIP | GGAGTTCCATGCGCCTTCGATATGACGATATTCGCTTTGGCA | 42 |
| | BIP | TGGTAACAAAGGCCATCGTCCCGCGGAATCAACTGGGGGAACCGGGA | 47 |
| | LF | AGCAACGTGGGAGCCAC | 17 |

**Notes:**
* Manually designed primer sets, Yr#1 is the selected primer set for the *Yr*-LAMP assay.
The underlined regions represent the complementary to 5′–3′ sequence.

(*Sambrook & Russell, 2006*). Insert transformation of *Y. ruckeri* (*pYr*), *Y. rohdei* (*pYro*) and *Y. frederiksenii* (*pYf*) plasmids was confirmed by colony PCR, using M13 forward and reverse primers (Invitrogen, Waltham, MA, USA) (Table S1). Transformed *E. coli* cells were grown on Luria-Bertani (LB) agar (1.5% (w/v) agar, 1% (w/v) tryptone, 1% (w/v) NaCl, 0.5% (w/v) yeast extract, pH 7.5), with the addition of 50 μg/ml of kanamycin to select for transformed cells. Plasmid isolations were performed using the FastGene® Plasmid Mini Kit (NIPPON Genetics Co. Ltd, Tokyo, Japan) and, eluted in 50 μl of elution buffer and stored at −20 °C. Total plasmid concentration was determined by Qubit™ 1x dsDNA BR Assay Kit (Invitrogen, Waltham, MA, USA) using the Qubit 4 Fluorometer (Thermo Fisher Scientific, Waltham, MA, USA).

### *Yersinia ruckeri* LAMP–*Yr*-LAMP

The four proposed *Yr*-LAMP primer sets were tested against 0.5 ng/μl of *pYr* initially using default primers concentration of 0.8 μM of FIP & BIP, 0.2 μM of F3 & B3 and 0.4 μM of LF and LB for Yr#2). All LAMP reaction volumes were 25 μl, consisting of 15 μl GspSSD2.0 Isothermal Mastermix (ISO-004; OptiGene, Horsham, England), 5 μl of primer mixture and 5 μl template (plasmid or genomic DNA extract). The template was replaced by TE buffer (10 Mm Tris-HCl and 0.1 mM ethylenediamine tetraacetic acid (EDTA), pH 8.0) in

no template control (NTC) samples. Reactions were performed using the Genie® II and Genie® III machines (OptiGene Limited, Horsham, England) initiated with a pre-heating step of 40 °C for 1 min, followed by 30 min of isothermal amplification at 65 °C then an annealing step where the temperature dropped from 94 °C to 84 °C with a 0.5 °C/s drop rate.

After obtaining the optimal primer set, a series of assays were conducted to determine the most effective primer concentration, varying the amounts of FIP, BIP, and LF. Reactions were performed and analysed using the Genie® II and Genie® III machines, as previously described, with 30 min of amplification. Primer concentrations were set at 0.2 μM for F3 and B3, 1.6 μM for FIP and BIP and 0.8 μM LF.

### Determination of sensitivity for *Yr*-LAMP

The assay's sensitivity was assessed using serial dilutions of *pYr*. The template was sequentially 10-fold diluted in TE buffer from 0.5 ng/μl until $0.5 \times 10^{-9}$ ng/μl and tested with optimised Yr#1 primer concentrations (*Keeling et al., 2012*). The reactions were repeated 10 times and analysed using the Genie® II machine, as previously described, with 30 min of amplification to assess the limit of detection (LOD).

### PCR for *Y. ruckeri*

Conventional PCR (cPCR) was employed to amplify a portion of the *glnA* target region associated with the *Yr*-LAMP assay, facilitating a sensitivity comparison. The amplification utilized specific primers for *Y. ruckeri* (*glnA*), designed by *Keeling et al. (2012)* (Table S1). The PCR reaction mixture comprised a total volume of 25 μl, containing a final concentration of 1X Promega GoTaq® Green Master Mix, 0.4 μM of each primer, and 5 μl of the template. The products of the PCR were analyzed through electrophoresis on a 1.5% (w/v) agarose gel, stained with 0.2 μg/mL ethidium bromide, and run at 100 v for 35 min to enable clear visualization of the amplification results.

### *Yr*-LAMP specificity testing

Initial specificity testing was performed using 0.25 ng/μl of target plasmid (*pYr*) and 100-fold higher concentration (25 ng/μl) of plasmids containing gene homologous from closely related species, *pYro* or *pYf*. Reactions were performed and analysed using the Genie® II and Genie® III machines, as previously described, with 30 min of amplification.

To further validate the test's specificity, the *Yr*-LAMP assay was evaluated against synthetic DNA representing 999 bp of *glnA* from salmon pathogens, *Aeromonas salmonicida* subsp. *Salmonicida*, *Flavobacterium psychrophilum*, and *Renibacterium salmoninarum*. This evaluation also included a range of other bacterial species. Total genomic DNA was extracted using Bioneer AccuPrep Genomic DNA extraction kit from a bacterial specificity panel consisting of gram-positive bacteria including, *Enterococcus faecalis*, *Staphylococcus aureus*, *Streptococcus agalactiae*, *S. pyogenes*, *S. salivarius* and *S. sanguinis* and gram-negative bacteria including, *Escherichia coli*, *Pseudomonas aeruginosa*, *P. fluorescens* and *Vibrio natriegens* species following manufacturer instructions. Total DNA concentrations were determined by Qubit™ 1× dsDNA BR Assay

Kit (Invitrogen, Waltham, MA, USA), using Qubit 4 Fluorometer (Thermo Fisher Scientific, Waltham, MA, USA) and were stored at −20 °C until needed.

Furthermore, both fresh and seawater samples were collected and screened. To assess the water collection contaminants, seawater samples and fresh lake water (La Trobe Lake, Victoria, Australia) were filtered using the filtration method outlined in Fig. 1B (*Khodaparast et al., 2022*). The released genomic DNA was used directly as LAMP template or stored at 4 °C until further use. The presence of bacteria in those concentrated water samples was confirmed using a universal bacterial PCR assay targeting the *16S* gene of both gram-positive and gram-negative bacteria (*Barghouthi, 2011*). The amplification was performed using a primer mixture of seven primers, Golden mixture 7 (Table S1). The PCR reaction mixture totalled 25 µl, consisting of a final concentration of 1X Promega GoTaq® Green Master Mix, 0.1 µM of each of the seven primers and 5 µl of concentrated seawater or lake water as a template. The PCR product was observed using 1.5% (w/v) agarose gel with 0.2 µg/mL ethidium bromide, with electrophoresis performed at 90 v for 70 min (*Barghouthi, 2011*).

LAMP was completed following the isolation of genomic DNA, plasmids containing gene homologous of closely related species, synthetic DNA of salmonid pathogens and the concentration of water samples. Using the optimised concentration of the selected *Yr*-LAMP primer set at 65 °C, the LAMP reaction was run for 30 min to detect the amplification. Reactions were performed in triplicates and tested by the Genie® II and Genie® III machines; as previously described, the annealing temperature of the assay was used to confirm the correct product.

## Isolation of DNA from water samples for in-field detection

A single colony of *E. coli* transformed with *pYr* was grown in 5 ml of LB broth at 37 °C with shaking at 220 rpm until reaching $OD_{600} = 1.0$, measured by the Halo DNAmaster (Dynamica). Cell numbers were calculated using the standardized equation of $OD_{600}$ of $1.0 = 8 \times 10^8$ cells/ml for *E. coli*. Environmental seawater was spiked with 10-fold serial dilutions of *E. coli* cells containing *pYr*. For each sample, 0.5 ml of cells was added to 45.5 ml of seawater (collected from St Kilda beach, Victoria, Australia) samples using the filtration method developed by *Khodaparast et al. (2022)*. Briefly, the water sample was filtered using a Target2™ GMF (Glass MicroFiber) Syringe Filters with a 1.2 µm pores size. Then, samples were refiltered through a polyethersulphone (PES) Whatman™ Uniflo™ Syringe Filters with a 0.45 µm pores size. The filter unit was washed with 5 ml of sterile water, and subsequently back flushed with 200 µl of sterile water into a 1 cc syringe. The bacterial cells in the recovered water from the filter were lysed using 0.3 M KOH (1:1 ratio) as a lysis buffer, and 5 µl of the mix was directly used in the *Yr*-LAMP reaction (Fig. 1).

## Data analysis

The co-efficient of variation (CV%) was calculated to indicate repeatability using the equation:

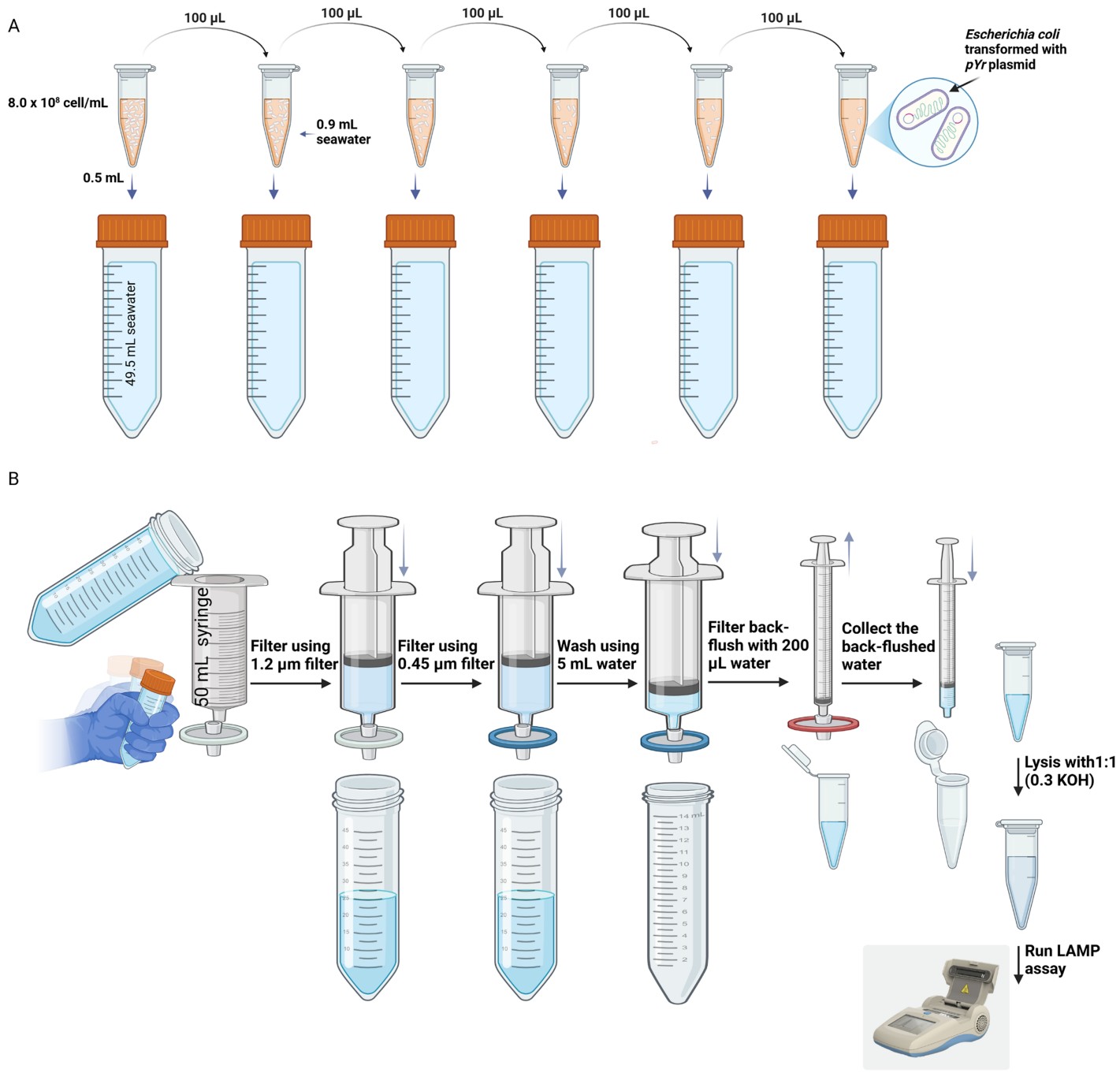

**Figure 1  Artificially constructed assay procedure.** Illustration of (A) the spiking and (B) filtration method for testing the sensitivity of the LAMP assay with contaminated seawater. 0.3M KOH was used as in-field lysis method at room temperature (R.T). MQ (Milli-Q) water. pYr is plasmid harbouring glnA from *Yersinia ruckeri*. Created in BioRender. Abbas, 2025; https://BioRender.com/v03z528.

CV% = (Standard deviation/mean) × 100. The sensitivity experiment was performed 10 times to assess the inter-assay variation (CV%). Primer optimisation, specificity and spiking experiments were repeated three times to confirm the results.

## RESULTS

### Primer screening

The target gene in this study, glutamine synthetase (*glnA*) from *Y. ruckeri*, was chosen for its divergent DNA sequence following alignment comparisons with similar genes within the national center of biotechnology information (NCBI) database. When manually identifying the optimal region of the *glnA* sequence for designing LAMP primers, an alignment of the *glnA* gene from *Y. ruckeri* revealed that the region between 70 and 585 bp showed over 99% similarity among *Y. ruckeri* strains and less than 90% similarity with various *Yersinia* species. Using the sequence mentioned above as a template for designing primers resulted in four primer sets with Yr#1 and Yr#2 software generated and Yr#3 and Yr#4 manually designed (Table 1). Manually designed LAMP primers allowed for greater flexibility in targeting specific areas of *glnA* that had lower percentage similarity. The Yr#1, Yr#2, Yr#3 and Yr#4 were designed to amplify regions with approximate similarity 88%, 89%, 90% and 89%, respectively, to non-target sequences. Initial testing of the four different primer sets for *Y. ruckeri-glnA* resulted in varying time to positive amplification (Tp), ranging from 05:37 ± (7 s) to 17:47 ± (92 s) min (Table 2, Fig. S1). Primer set Yr#1, which consists of five primers designed to amplify 209 bp region from 219 to 427 bp of the *Y. ruckeri glnA* gene (Fig. 2), gave the fastest Tp (05:37 min ± 7 s) and was chosen for all future experiments.

### Optimisation of *Yr*-LAMP primers

Manipulation of the LAMP reaction conditions were required to improve the Tp and the assay sensitivity. Initially, performance was tested using different concentrations of inner, outer and loop primers of Yr#1 (Table 3). Seven different concentrations of primers within Yr#1 set were tested against 0.5 ng/μl of *pYr* and, four different combinations of primer concentrations resulted in average Tp less than 5:42 min (Table 3). Those four primer combinations were assessed using 0.25 ng/μl of the *pYr* and 100-fold greater concentration of closely related *Yersinia* species, *Y. rohedi* and *Y. frederiksenii*. The primer concentration of 1.6, 0.2, 0.8 μM for FIP & BIP, F3 & B3 and LF respectively, showed the highest sensitivity and specificity, with a Tp of 05:27 min ± (12 s) for *pYr*, no detection of *pYro* and a Tp of 24:00 min ± (0 s) for *pYf* (Table 4). The amplification of the higher concentration of *Y. frederiksenii* plasmid was observed at 24 min, exceeding the 20-min cutoff time specified by the LAMP Mastermix manufacturer. Therefore, this primer set concentration was deemed specific to *Y. ruckeri* (Table 4). Ultimately, 1.6, 0.2, 0.8 μM of FIP & BIP, F3 & B3 and LF primers, respectively, of Yr#1 set were chosen to be the optimal *Yr*-LAMP concentrations to discriminate between *Y. ruckeri* and closely related *Yersinia* species.

**Table 2 Amplification times for the four suggested LAMP primer sets with their time to positive (Tp) and standard deviation values.**

| Primers set | Yr#1* | Yr#2 | Yr#3 | Yr#4 |
|---|---|---|---|---|
| Tp (mm:ss) | 05:37 ± 7 s | 06:30 ± 15 s | 07:42 ± 80 s | 17:47 ± 92 s |

**Note:**
* Primer set selected to be used throughout *Yr*-LAMP study.

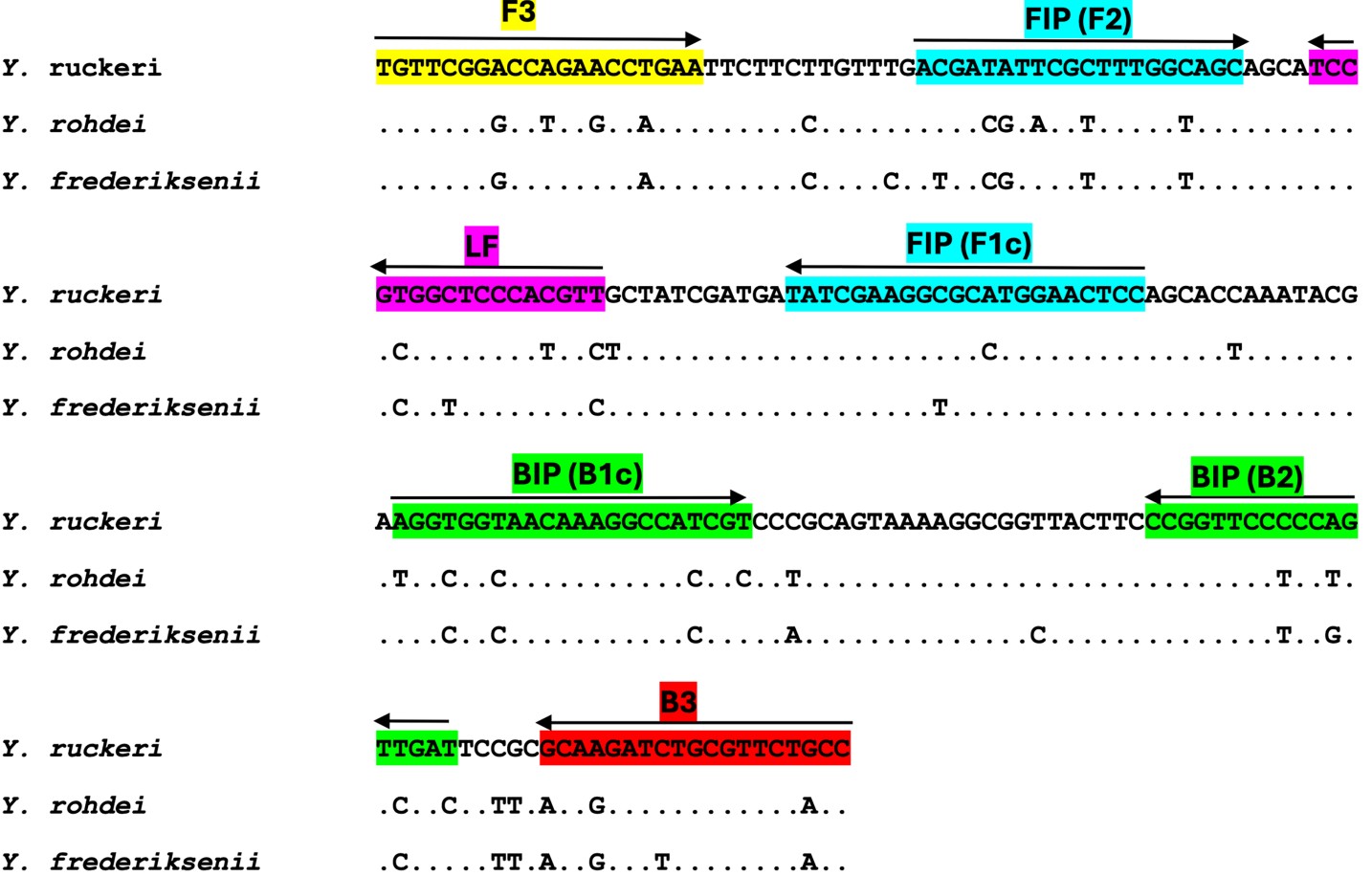

**Figure 2 Pairwise dots alignment of the target region of *Yersinia ruckeri* primer set 1 (Yr#1) targeting the glutamine synthetase gene (glnA) (209 bp) and the corresponding sequences of the closely related species *Yersinia rohedi* and *Yersinia frederiksenii*.** The Yr#1 primer set consists of two outer primers, F3, B3, two inner primers FIP (F2+F1c), BIP (B2+B1c) and forward loop primer, LF which are highlighted to show the regions of annealing with the template. Arrows indicate the direction of the amplification.

## Specificity assessment of *Yr*-LAMP

Further specificity was performed using an optimized primer set against a range of gram-positive and gram-negative bacteria (Fig. 3). Presence of bacteria in fresh and seawater samples was confirmed by PCR using universal bacterial primers targeting *16S* gene. Agarose gel then confirmed that bacteria were present in the filtered water sample, as a multiple band profile was observed (Fig. 4). Seawater sample #8 and lake water sample #4 were selected to resemble the environmental water in further experiments. A single amplification peak specific to *pYr* amplification was detected approximately at 6 min with

**Table 3 Initial screening of primer set Yr#1 concentrations.**

| FIP & BIP, F3 & B3, LF (μM) | Average Tp (mm:ss) |
|---|---|
| **0.8, 0.2, 0.4** | **05:13 ± 04 s** |
| 1.2, 0.2, 0.2 | 07:02 ± 05 s |
| **1.2, 0.2, 0.4** | **05:15 ± 12 s** |
| **1.2, 0.2, 0.8** | **05:42 ± 05 s** |
| 1.6, 0.2, 0.2 | 07:15 ± 00 s |
| 1.6, 0.2, 0.4 | 06:15 ± 00 s |
| **1.6, 0.2, 0.8** | **05:25 ± 07 s** |

Note:
Concentrations in bold were selected for further assessment.

**Table 4 Final assessment of optimum concentration of primer set Yr#1 primers.**

| FIP & BIP, F3 & B3, LF (μM) | Average Tp (mm:ss) | | |
|---|---|---|---|
| | *Y. ruckeri* (0.25 ng/μl) | *Y. rohedi* (25 ng/μl) | *Y. frederiksenii* (25 ng/μl) |
| 0.8, 0.2, 0.4 | 04:54 ± 05 s | 22:37 ± 63 s | 18:09 ± 37 s |
| 1.2, 0.2, 0.4 | 05:29 ± 40 s | 24:49 ± 288 s | 16:24 ± 195 s |
| 1.2, 0.2, 0.8 | 05:36 ± 15 s | 20:08 ± 411 s | 15:20 ± 93 s |
| **1.6, 0.2, 0.8** | **05:27 ± 12 s** | **Not detected** | **24:00 ± 0 s** |

Note:
Concentrations in bold were selected to be used in all further *Yr*-LAMP assays.

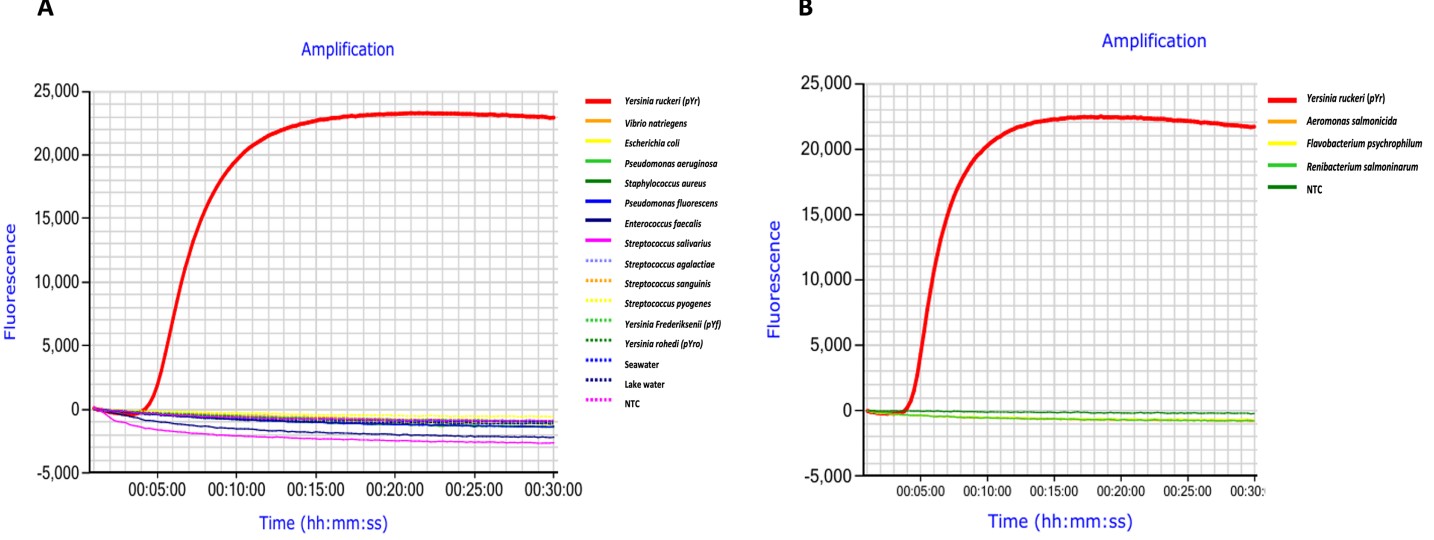

**Figure 3 Specificity assessment of *Yersinia ruckeri* (Yr-LAMP) assay.** The figure illustrates the real-time LAMP results, and the fluorescent curve represents amplification. Optimised Yr-LAMP conditions were tested against 0.5 ng/μl of plasmid (pYr) containing the target gene, glutamine synthetase (glnA) of *Yersinia ruckeri* against (A) whole genomic DNA extract of gram-positive and gram-negative bacteria, plasmid harbouring glnA of related bacteria *Yersinia rohde i* (pYro) and *Yersinia frederiksenii* (pYf), seawater or lake water extracts and (B) gene blocks of glnA of other salmonid pathogens. TE buffer replaced the template DNA in the no template control (NTC).

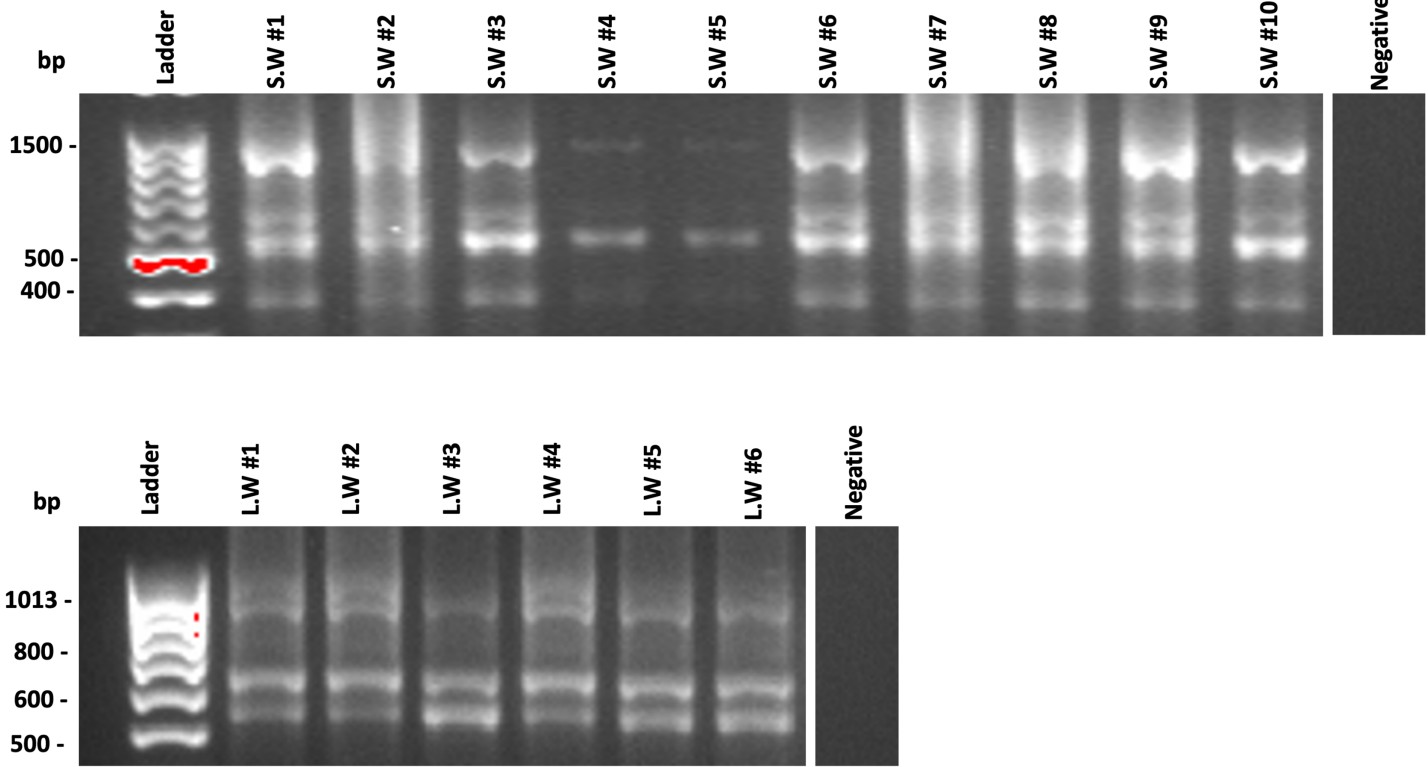

**Figure 4 Universal bacterial detection assay for different samples of filtered (A) seawater (S.W) or (B) lake water (L.W).** Five μl of conventional polymerase chain reaction (cPCR) of universal bacterial assay product was loaded to 1.5% (w/v) agarose gel with 0.2 μg/mL ethidium bromide and resolved for 35 min at 100 v. TE buffer replaced the template DNA in negative control samples.

Tm 88.2 °C. There was no amplification for any of the non-target samples during the 30 min LAMP amplification time (Fig. 3), indicating that the *Yr*-LAMP assay was specific for *Y. ruckeri* and did not display cross-reactivity with any of the other panel bacteria or other microorganisms which naturally inhabit seawater or lake water.

## Analytical performance of *Yr*-LAMP assay

LAMP sensitivity was evaluated using a 10-fold serial dilution starting at approximately $1.13 \times 10^8$ copies/μl of the *Y. ruck*eri *glnA* gene. The optimized *Yr*-LAMP assay conditions could successfully detect as low as $1.13 \times 10$ copies/μl of *pYr* in less than 20 min (Table 5, Fig. S2). All amplicons showed a consistent melting temperature of approximately 88 °C, even at a concentration of $0.5 \times 10^{-8}$ ng/μl. However, this low concentration produced unreliable Tp results, with inconsistent amplification across assays and a CV% greater than 15%. Excluding the $0.5 \times 10^{-8}$ ng/μl dilution, the maximum inter-assay CV% for the sensitivity tests was 9.9%. Based on the sensitivity test results, a sample is considered positive if amplification occurs within 20 min with an inter-assay CV% of ≤15%. Consequently, any subsequent fluorescent peaks are deemed negative. The corresponding cPCR could only amplify the 109 bp specific to the target sequence till the $0.5 \times 10^{-4}$ ng/μl equivalent to $1.13 \times 10^4$ copies/μl, three-folds less sensitive than *Yr*-LAMP assay (Table 5, Fig. S3).

**Table 5** *Yersinia ruckeri* (*Yr*-LAMP) assay and conventional PCR (cPCR) limit of detection (LOD).

| Copy number (copies/µl) | Plasmid concentration (ng/µL) | Average Tp (mm:ss) | Inter-assay CV% | Conventional PCR |
|---|---|---|---|---|
| $1.13 \times 10^8$ | 0.5 | 05:51 ± 33 s | 9.4 | Positive |
| $1.13 \times 10^7$ | $0.5 \times 10^{-1}$ | 06:24 ± 16 s | 4.3 | Positive |
| $1.13 \times 10^6$ | $0.5 \times 10^{-2}$ | 07:25 ± 26 s | 5.8 | Positive |
| $1.13 \times 10^5$ | $0.5 \times 10^{-3}$ | 08:28 ± 39 s | 7.8 | Positive |
| $1.13 \times 10^4$ | $0.5 \times 10^{-4}$ | 09:37 ± 25 s | 4.4 | Positive |
| $1.13 \times 10^3$ | $0.5 \times 10^{-5}$ | 11:20 ± 46 s | 6.8 | Negative |
| $1.13 \times 10^2$ | $0.5 \times 10^{-6}$ | 13:37 ± 64 s | 7.9 | Negative |
| $1.13 \times 10$ | $0.5 \times 10^{-7}$ | 15:52 ± 119 s | 12.5 | Negative |
| 1.13 | $0.5 \times 10^{-8}$ | 22:08 ± 299 s | 22.5 | Negative |
| $1.13 \times 10^{-1}$ | $0.5 \times 10^{-9}$ | Not detected | Not detected | Negative |

### Environmental sampling performance of *Yr*-LAMP assay

Seawater known to be free from *Y. ruckeri* was collected from St. Kilda beach and used to assess a filtration and simple extraction method suitable for field-use and LAMP detection. Each sample containing either 50 ml of water was spiked with decreasing quantities of *E. coli* cells containing the *glnA* gene of *Y. ruckeri*, or a negative control of unspiked seawater, was filtered and extracted with an equal volume of 0.3 M KOH. *Yr*-LAMP confirmed detection to the equivalent of 0.08 cells/µl within 14 min with a melting temperature of approximately 88 °C (Table 6, Fig. S4). However, the corresponding cPCR detected only 8.0 cells/µl (Table 6, Fig. S5).

### DISCUSSION

In this study, we utilized competent *E. coli* cells integrated with synthetic DNA encoding the *Y. ruckeri* glutamine synthetase gene for DNA extraction and sensitivity testing. The use of *E. coli* was necessitated by the unavailability of *Y. ruckeri* cultures or clinically infected samples. Our DNA extraction method, which employed KOH, proved effective even on gram-positive bacteria, indicating its robustness and potential applicability across different bacterial types (*Ackerly, Tran & Beddoe, 2024*). Initial assessments demonstrated that while *E. coli* served as a useful surrogate due to its gram-negative nature, it is imperative to validate these findings with *Y. ruckeri* bacterial cells to ensure accurate sensitivity values. To advance this validation, efforts should be made to acquire *Y. ruckeri* samples and replicate the DNA extraction and sensitivity tests on *Y. ruckeri* cells and compare the results with those derived from *E. coli*.

While the *Yersinia* genus encompasses a range of species, including *Y. enterocolitica* and *Y. frederiksenii*, some of which have been found in fish, not all are associated with fish diseases (*Carson et al., 2008*). Therefore, it is essential to identify a distinct target gene that ensures high specificity for detecting *Y. ruckeri* amidst other related and unrelated bacteria present in the fish environment (*Carson et al., 2008*). The *16S rRNA* gene, which is approximately 1,500 base pairs in size, is one of the most conserved genes in bacteria due to its critical role in cell function. It is also the most extensively represented gene in the

**Table 6 Average time-to-positive (Tp, mm:ss) values of seawater samples spiked with *E. coli* cells harbouring *pYr* by LAMP.**

| Concentration (cells/µl) | Average Tp (mm:ss) | Average Tm (°C) | Conventional PCR |
|---|---|---|---|
| 8,000 | 5:29 ± 13 s | 87.8 | Positive |
| 800 | 6:13 ± 5 s | 87.8 | Positive |
| 80 | 7:54 ± 24 s | 87.6 | Positive |
| 8 | 8:56 ± 21 s | 87.6 | Positive |
| 0.8 | 10:45 ± 37 s | 87.8 | Negative |
| 0.08 | 13:44 ± 42 s | 87.6 | Negative |
| Positive control | 5:41 ± 5 s | 88.2 | Positive |

GenBank database, with over 90,000 sequences available (*Clarridge, 2004*; *Hao, Pei & Brown, 2017*; *Janda & Abbott, 2007*). The aforementioned reasons make *16S rRNA* the ideal candidate for designing specific PCR primers however, challenges may occur when determining eight distinct regions for designing LAMP primers (*Carson et al., 2008*; *Notomi et al., 2000*). The *glnA* gene is a promising marker for identifying *Y. ruckeri* because it is conserved among various strains. A real-time PCR assay targeting the *glnA* gene of *Y. ruckeri* demonstrated 100% specificity in detecting ERM infection (*Keeling et al., 2012*; *Kotetishvili et al., 2005*). The conservation of this gene sequence within *Y. ruckeri* makes the assay a versatile diagnostic tool for identifying new bacterial variants globally (*Kotetishvili et al., 2005*).

The previously published ERM-LAMP assay utilized five primers targeting the *yruI/yruR* gene of *Y. ruckeri* in tissue samples. While the assay demonstrated high specificity for *Y. ruckeri*, it required one hour for amplification (*Saleh, Soliman & El-Matbouli, 2008*). The newly developed *Yr*-LAMP assay comprises five LAMP primers targeting the *glnA* gene of *Y. ruckeri*, which was conserved in all tested *Y. ruckeri* strains and varied from other species within the same genus (*Kotetishvili et al., 2005*). The assay was optimised to use primer concentrations of 1.6, 0.2, 0.8 µM of FIP & BIP, F3 & B3 and LF, respectively, of the Yr#1 primer set to achieve high specificity and sensitivity. All the amplicons had a similar melting temperature of approximately 88 °C, even at a concentration of $0.5 \times 10^{-8}$ ng/µl. However, this concentration resulted in stochastic amplification and was therefore considered inconsistent and excluded. While the existing ERM-LAMP and real time PCR has analytical sensitivity of 1 pg and 5 fg respectively, our newly developed *Yr*-LAMP assay can detect $0.5 \times 10^{-7}$ ng/µl of plasmid DNA which is equivalent to 11.3 copies/µl in under 20 min. This is a shorter detection time compared to both of those assays (*Keeling et al., 2012*; *Saleh, Soliman & El-Matbouli, 2008*). Accurate sensitivity comparison among the three assays is not possible since our assay was tested on plasmids with the target gene, while the others used genomic DNA. Nonetheless, our *Yr*-LAMP assay is highly sensitive, three-fold more sensitive than the corresponding cPCR and with a detection limit comparable to other published LAMP assays (*Agarwal et al., 2022*).

Previous studies tracked the route of *Y. ruckeri* in the host fish, the bacteria enter the fish *via* gills and shortly it reaches the intestine although it was not observed in kidney until the

third day post infection and a week later was observed in the brain and other internal organs (*Méndez & Guijarro, 2013*; *Ohtani et al., 2014*). Those findings support the concept of early detection of pathogen is a key factor for containing infection and mitigating the excess use of antimicrobial treatment (*Yu et al., 2022*). Sampling and extraction can be considered the most critical steps in diagnostics, they can promote the assay to be field-deployable or not. The current assays for detecting *Y. ruckeri* are either invasive using fish tissues (*Keeling et al., 2012*; *Saleh, Soliman & El-Matbouli, 2008*) or non-invasive using faeces (*Carson et al., 2008*; *Ghosh et al., 2018*) or blood samples (*Altinok, Grizzle & Liu, 2001*; *Bastardo, Ravelo & Romalde, 2012*). Although the published *Y. ruckeri* detection methods are rapid (approximately one hour) for LAMP and real time PCR, the extraction step itself was long. Extracting genomic DNA from tissue samples, blood, or faecal samples with commercial extraction kits required several hours and purifying them from PCR inhibitors took even longer. Therefore, to address these challenges for our *Yr*-LAMP assay application, we developed field-tolerant sampling and extraction methods from environmental water (*Carson et al., 2008*; *Keeling et al., 2012*; *Saleh, Soliman & El-Matbouli, 2008*). For testing the field performance of *Yr*-LAMP assay, environmental water spiked with serial dilutions of *E. coli* bearing the target gene was filtered and lysed using KOH. The *Yr*-LAMP assay could successfully detect as low as 0.08 cells/µl ($3.9 \times 10^{-10}$ ng/µl of *pYr*) of the initial collected water in less than 15 min. Relating these results to the sensitivity test, we can suggest that approximately $1.13 \times 10^2$ copies/µl of *pYr* were recovered and extracted from the initial 4,000 cells spiked into the 50 ml water. On the other hand, cPCR could only detect until 8.0 cells/µl of the initial 50 ml water sample, two-fold less sensitive than the *Yr*-LAMP assay.

It was confirmed that *Yr*-LAMP only amplified the positive control and no other bacteria, suggesting that *Yr*-LAMP is specific. As the assay will be used on environmental samples there is potential for cross reactivity with preexisting microorganisms present in the water habitat, water samples from the beach and lake were collected, filtered, and concentrated using the filtering method. Although a diverse range of microorganisms exists in both freshwater and seawater, the microbiome extracted from these samples did not result in amplification by our developed LAMP assay. This outcome demonstrates the assay's high specificity (*Baharum et al., 2010*; *Wang et al., 2012*). Although using KOH for direct extraction of DNA in-field was successful and sensitive in previous work (*Ackerly, Tran & Beddoe, 2024*; *Khodaparast et al., 2022*), it was observed to have a slight inhibitory effect on LAMP represented in a minor decline in Tm than the positive control. Those observations are consistent with other findings regarding the effect of inhibitors on the melting temperature of LAMP (*Nwe, Jangpromma & Taemaitree, 2024*; *Shirshikov & Bespyatykh, 2022*). However, despite this effect on Tm, the inhibition did not impact the assay's sensitivity, and therefore it was not considered problematic.

Studies have demonstrated that *Y. ruckeri* can survive for up to 90 days or more in both fresh and marine waters (*Romalde et al., 1994*). Therefore, detecting *Y. ruckeri* in water samples can inform real-time management strategies. For instance, the presence of *Y. ruckeri* in sea cages may indicate the need for treatment or the delay of introducing new fish into the sea cages.

The data reported in our *Yr*-LAMP assay was obtained by using real-time LAMP instruments, Genie®ll or Genie®lll (OptiGene, Horsham, England) real time fluorometer. Genie is a lightweight, portable device that can work with batteries. Despite the widespread use of the Genie instruments in numerous studies globally (*Enicks, Bomberger & Amiri, 2020*; *Fowler et al., 2021*; *Neeraja et al., 2015*; *Saar et al., 2021*), they are constrained by the limited number of chambers, with Genie®lll offering eight and Genie®ll providing 16 wells. This restricts the number of samples available for testing to six and 14, respectively, excluding positive and negative controls. Furthermore, the capacity is reduced by half if duplicate tests are necessary, making these instruments unsuitable for high-throughput detection. For wide range surveillance, other detection alternatives can be used, such as lateral flow dipstick (LFD) or visualise colour detection. For example, *Yu et al. (2022)* developed an impressive LAMP assay capable of detecting Singapore grouper iridovirus (SGIV) directly from fin samples that were boiled in 50 µl of 0.02 N NaOH. The results of the LAMP assay were indicated by a colour change from yellow to pink, with the entire process taking approximately an hour and was able to detect less than 6 copies/µl of the virus. The straightforward extraction process combined with the rapid and sensitive nature of the assay makes it an effective on-site method for efficiently managing SGIV infections (*Yu et al., 2022*).

LFD has gained recent popularity due to its versatility in detecting various pathogens, including bacteria (*Thongkao et al., 2015*), virus (*Ding et al., 2010*) and parasites (*Ding et al., 2010*). Its ease of use, exemplified by the accessibility of COVID-19 test strips even for non-specialists, underscores its practicality (*Yüce, Filiztekin & Özkaya, 2021*). Looking ahead, exploring the feasibility of adapting the *Yr*-LAMP assay into an LFD format presents an opportunity to circumvent the testing limitations inherent in Genie devices. By integrating swabbing or the straightforward filtration and extraction methods demonstrated in our study, we aim to develop a high-capacity, device-free, point-of-care detection method suitable for deployment by untrained personnel.

## CONCLUSIONS

*Yersinia ruckeri* poses significant economic risks in the aquaculture industry, necessitating regular surveillance to pre-emptively manage infections and reduce reliance on antimicrobial agents, thereby preventing outbreaks. The *Yr*-LAMP primers exhibit high specificity, effectively discriminating against organisms naturally occurring in fish environments and closely related *Yersinia* species. This method identifies the pathogen at extremely low levels, with an analytical sensitivity of 0.08 cells/µl in less than 20 min. Integrating these specific LAMP primers with filtration and KOH extraction enhances the system's speed, reliability, affordability, and simplicity, making it feasible for farm workers to detect *Y. ruckeri* infections early and effectively control outbreaks.

## ACKNOWLEDGEMENTS

We would like to thank Mr. Lawrence Liversage for providing the bacteria used in the specificity screening.

### Funding

This research was supported by the Cooperative Research Centres Project (CRC-P), which was awarded to GeneWorks and La Trobe University. The funders had no role in study design, data collection and analysis, decision to publish, or preparation of the manuscript.

### Grant Disclosures

The following grant information was disclosed by the authors:
Cooperative Research Centres Project (CRC-P).
GeneWorks and La Trobe University.

### Competing Interests

Travis Beddoe is an Academic Editor for PeerJ.

### Author Contributions

- Hoda Abbas conceived and designed the experiments, performed the experiments, analyzed the data, prepared figures and/or tables, authored or reviewed drafts of the article, and approved the final draft.
- Nickala Best conceived and designed the experiments, authored or reviewed drafts of the article, and approved the final draft.
- Gemma Zerna analyzed the data, authored or reviewed drafts of the article, and approved the final draft.
- Travis Beddoe conceived and designed the experiments, analyzed the data, authored or reviewed drafts of the article, and approved the final draft.

### Data Availability

The raw LAMP and DNA agarose gel data is shown for each experiment.

### Supplemental Information

Supplemental information for this article can be found online at http://dx.doi.org/10.7717/peerj.19015#supplemental-information.

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
