# Peer review of "Development of LAMP assay for early detection of Yersinia ruckeri in aquaculture"

_PeerJ, doi:10.7717/peerj.19015_

## Round 0.1 · original submission · Major Revisions

Please revise the manuscript by referring to the reviewers' comments. A point-by-point response letter is recommended to state how the authors respond to the queries and comments and where the manuscript is revised accordingly. We look forward to receiving the revised version of your manuscript.

Reviewer 1 ·

Basic reporting

The study design is well explained and the results are supported by the study design. Nevertheless, it is recommended that the authors attend to a few comments outlined below.

Experimental design

The authors must include other important bacterial pathogens in the present study to make more specific for developed LAMP such as Aeromonas salmonicida, Flavobacterium psychrophilum and Renibacterium salmoninarum. These species are equally importance to include in the LAMP. Please repeat the experiment.

Validity of the findings

Please clarify in detail, why glutamine synthetase (glnA) gene was selected for LAMP but not other genes.
Figures 1 and 2: The quality of the figures needs to be updated and enhanced.

Additional comments

Figure 1: add legend in more details.
Supplemental Table 1: Please move this table to main document.
Supplemental Figure 1 and Figure 3: Please move these supplementary Figures to main document. Additionally, the quality of the figures needs to be updated and enhanced.

Reviewer 2 ·

Basic reporting

The manuscript written in a clear language but the punctuation needs to be improved and there are quite few language-related deficits. Figures and tables are clear. The reference list is missing at least one reference (Stentiford et al., 2017).

Experimental design

No comments

Validity of the findings

The results are based on solid experimental design and the findings are novel and relevant.

Additional comments

No comments

Reviewer 3 ·

Basic reporting

The article has been written in professional English. Sometimes more information can be included e.g. in the introduction where more information on LAMP is needed. The discussion needs to address better that the assay has not been tested on real Yersinia samples and what a positive result from water samples means related to diseased fish.

Experimental design

Abbas et al. developed a field-deployable molecular assay using loop mediated isothermal amplification (LAMP) for the detection of Y. ruckeri, which causes severe diseases within salmonid aquaculture. The study was well designed including cross-reactivity and LOD testing and spiked field samples were included. A weakness is that the assay has not been tested on DNA derived from Yersinia cultures. As particularly the in-field extraction step is highlighted as one of this studies’ achievements, it has not been proven on real cell cultures. This should be discussed properly and highlighted for further research (it has been mentioned in L401-408 but should be done earlier and in more detail). Additionally, it needs to be discussed how finding Yersinia in the water concludes to diseased fish as it is known to universally appear in the aquatic environment.

Validity of the findings

As mentioned above the missing test on real cultures is a weakness that needs to be addressed better.

Additional comments

Abstract
L50: It would be interesting to compare the LOD to common PCR assays here to have reference values.
L51: Here it is unclear that E.coli was just a vector for the Yersinia Gblocks. Please include more detail.
Introduction
L77: add “in” salmonids.
L117: The technology and mechanism behind LAMP should be covered a bit mor in detail here.
L124: Give more details about the established LAMP assays and how they have been applied.
L125: What does ERM stand for and again give more details on this test including target region, primers and LODs.
Material and Methods
L137: it is not clear to me if and why there have been new primers designed for Yersinia ruckeri when there was already a valid LAMP test out there (ERM) as described in the introduction?
L149: also known as Gblocks or plasmids? Rephrase “constructs”. And I assume these have also been constructed for cross-reactivity testing? Please add this.
L155: “pYr/h/fglnA…” these abbreviations are quite hard for the reader to keep in mind and differentiate.
Results
L260: add positive “amplification”.
L267-271: Is repetitive and could be combine to simply state the most optimal conditions. And then start with L273.
L276: Needs to be rephrased and not start with “despite” but report the cross-amplification in detail.
L283-288: Is repetitive to M&M and/or needs to move to Discussion.
Discussion
Despite a very thorough experimental set-up, the weakness of the study consists in not having tested the assay on real cultured Yersinia strains. This should be discussed in the beginning and in detail.
Until L329 its repetition of the introduction and can be removed.
L342: Here the interesting aspect starts in what these challenges of 16S are? I would recommend starting the discussion from L349.
L386: This value should be reported in ng/ul.
L389: Were these bacteria actually detected in detail through metabarcoding or just confirmed by a general band on the gel? If just in general, it cant be reported like this.
Supplementary Figure 1 should be in the main manuscript as it helps a lot with understanding the experimental design.
Table 5: correct to “Escherichia coli” and generally avoid abbreviations in the Figure and Table captions.
Figure 1: Display the sequences not as letters but as dots when all aligned nucleotides are the same and use letters when the alignment shows differences.

---

## Round 0.2 · accepted · Accept

The manuscript can be accepted now.

Reviewer 1 ·

Basic reporting

The authors have satisfactorily addressed all the reviewers' comments.

Experimental design

no comment

Validity of the findings

no comment

Additional comments

no comment

Reviewer 2 ·

Basic reporting

No comment

Experimental design

No comment

Validity of the findings

No comment

Additional comments

The authors have thoroughly revised the manuscript and addressed all the comments. I have no additional comments